# Photoinduced site-selective alkenylation of alkanes and aldehydes with aryl alkenes

Hui Cao[1,2], Yulong Kuang [1], Xiangcheng Shi[1], Koi Lin Wong[1], Boon Beng Tan[1], Jeric Mun Chung Kwan[1], Xiaogang Liu [1,2] & Jie Wu [1,2✉]

The dehydrogenative alkenylation of C-H bonds with alkenes represents an atom- and step-economical approach for olefin synthesis and molecular editing. Site-selective alkenylation of alkanes and aldehydes with the C-H substrate as the limiting reagent holds significant synthetic value. We herein report a photocatalytic method for the direct alkenylation of alkanes and aldehydes with aryl alkenes in the absence of any external oxidant. A diverse range of commodity feedstocks and pharmaceutical compounds are smoothly alkenylated in useful yields with the C-H partner as the limiting reagent. The late-stage alkenylation of complex molecules occurs with high levels of site selectivity for sterically accessible and electron-rich C-H bonds. This strategy relies on the synergistic combination of direct hydrogen atom transfer photocatalysis with cobaloxime-mediated hydrogen-evolution cross-coupling, which promises to inspire additional perspectives for selective C-H functionalizations in a green manner.

[1] Department of Chemistry, National University of Singapore, 3 Science Drive 3, Singapore 117543, Republic of Singapore. [2] National University of Singapore (Suzhou) Research Institute, 377 Lin Quan Street, Suzhou Industrial Park, Suzhou, Jiangsu 215123, P. R. China. ✉email: chmjie@nus.edu.sg

Olefins play an essential role in organic synthesis owing to their extremely rich chemistry[1]. The direct alkenylation of simple C–H bonds with alkenes is one of the most appealing approaches for the construction and derivatization of complex molecules[2,3]. While transition-metal-catalyzed alkenylations of arenes and heteroarenes have been extensively studied[4–6], analogous transformations of alkanes and aldehydes remain rare[7–20]. Expanding direct alkenylation to $C_{sp3}$–H and $C_{sp2}$(O)–H bonds is challenging due to the slow metal-mediated C–H cleavage[21], the instability of alkyl/acyl metallic intermediates[22,23], and the selectivity issues arising from the ubiquity of $C_{sp3}$–H bonds in organic molecules[24].

Transition-metal-catalyzed dehydrogenative alkenylation of alkanes and aldehydes through an auxiliary-assisted metal insertion was recently disclosed (Fig. 1a)[7–13]. However, these transformations usually require stoichiometric Cu$^{II}$ or Ag$^{I}$ salts as oxidants. Moreover, the heteroatom-containing directing groups can undergo intramolecular Michael addition and give cyclized products instead of the desired alkenylated products[7–13]. In 2019, Gevorgyan and coworkers disclosed an elegant radical relay strategy for the alkenylation of alcohols with visible-light-mediated palladium catalysis[14]. Nonetheless, the requirement of a silicon-based auxiliary lowers the atom economy of this strategy, and the substrate scope is limited to alcohols. On the other hand, auxiliary-free dehydrogenative alkenylation offers practical advantages and has attracted substantial interest (Fig. 1b)[15–20]. Although much progress has been achieved, the developed systems usually require stoichiometric peroxides and a large excess of the C–H partner (over 40 equivalents of the alkanes), and the substrate scopes are limited to simple hydrocarbons and aromatic aldehydes. In pursuing methods for olefin construction[25–27] and photomediated C–H bond functionalization[28–31], we aspired to develop an auxiliary-free and oxidant-free strategy for $C_{sp3}$–H and $C_{sp2}$(O)–H alkenylation with the C–H substrate as the limiting reagent (Fig. 1c). Importantly, this would allow the late-stage functionalization of advanced synthetic intermediates and pharmaceutical relevant molecules without de novo synthesis[32].

In this context, we were inspired by direct hydrogen atom transfer (HAT) photocatalysts that could achieve the straightforward activation of C–H bonds[33]. Of particular interest was the

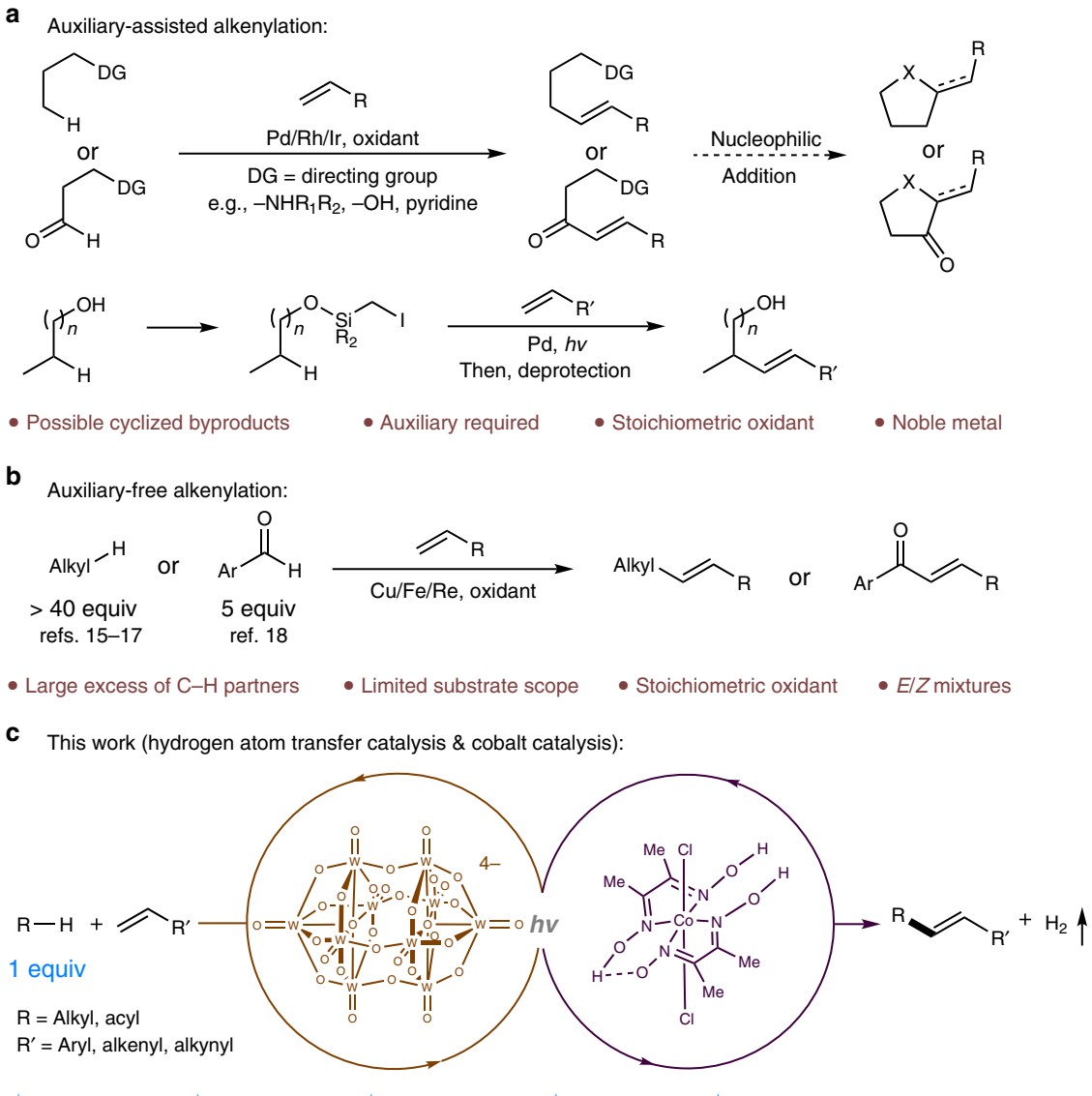

**Fig. 1 Strategies for the dehydrogenative alkenylation of alkanes and aldehydes with alkenes. a** Auxiliary-assisted alkenylation. **b** Auxiliary-free alkenylation. **c** Merging of a hydrogen atom transfer photocatalyst and a cobaloxime catalyst enables site-selective direct alkenylation of C–H substrates.

**Fig. 2 Proposed mechanism for the dehydrogenative alkenylation of alkanes and aldehydes.** The plausible mechanism involves a hydrogen atom transfer process, a single electron transfer process, and cobalt-catalyzed alkene formation and hydrogen evolution.

decatungstate anion ($[W_{10}O_{32}]^{4-}$), a polyoxometalate photo-catalyst that has been broadly applied in various functionalizations of alkanes and aldehydes[34–39]. On the other hand, photocatalytic dehydrogenative cross-coupling reactions with concomitant hydrogen evolution have recently been developed through cooperative photoredox and cobaloxime catalysis, as pioneered by the Wu and Lei groups[40–44]. Inspired by these studies, we proposed that the combination of decatungstate anion and a cobaloxime catalyst could enable the direct activation and alkenylation of alkanes and aldehydes using alkenes as the feedstocks. We herein report that by incorporating decatungstate catalysis and cobaloxime catalysis, direct alkenylation of alkanes and aldehydes with aryl alkenes can be achieved in the absence of hydrogen acceptors. This strategy features a broad substrate scope, high C–H site selectivity, excellent $E$ selectivity of the alkene products, and the use of the C–H substrate as the limiting reagent. Application of this strategy in late-stage functionalization was demonstrated by the selective alkenylation of natural products and pharmaceutically important molecules.

## Results

**Proposed reaction mechanism.** A hypothesized catalytic cycle is depicted in Fig. 2. Photoexcitation of decatungstate anion **1** would produce triplet excited state **2** after intersystem crossing[45]. Excited **2** can abstract hydrogen atoms from the alkane or aldehyde and afford carbon-centered radical R• and reduced decatungstate **3**[45]. Subsequent addition of radical R• to an alkene would furnish radical intermediate **4**, which can be reversibly captured by Co[II] **5** to form Co[III] intermediate **6**[46,47]. Photo-irradiation of the alkyl Co[III] complex would deliver the alkene product along with a Co[III]–H intermediate via a formal $\beta$-H elimination process[46,47]. Co[III]–H **7** would react with another proton to release $H_2$ and deliver Co[III] **8**[48–50]. Finally, a single-electron reduction of Co[III] **8** ($E_{1/2}$ Co[III]/Co[II] = −0.68 V versus Ag/Ag$^+$ in MeCN)[50,51] by reduced decatungstate **3** ($E_{1/2}$ $[W_{10}O_{32}]^{4-}/[W_{10}O_{32}]^{5-}$ = −0.96 V versus Ag/Ag$^+$ in MeCN)[52] would regenerate both catalysts. With this hypothesis in mind, we investigated the direct alkenylation of alkanes and aldehydes under various conditions.

**Reaction optimization.** Our investigation began with the alkenylation of cyclooctane with cyclooctane as the limiting reagent and styrene as the alkene partner. After extensive evaluation (Table 1 and Supplementary Tables 1–10), we established that the combination of tetra-$n$-butylammonium decatungstate (TBADT, **9**, 4 mol%), Co(dmgH)(dmgH)$_2$Cl$_2$ (**10**, 1 mol%), and 2,6-lutidine

(**L5**, 10 mol%) in acetonitrile (0.1 M) at 60 °C were the optimal conditions, affording alkenylated product **11** in 69% isolated yield with exclusive $E$ selectivity (Table 1, entry 1). Reducing the amount of styrene from 10 equiv. to 5 equiv. resulted in a much lower yield (Table 1, entry 2). After careful analysis of the crude product mixture using GC-MS and $^1H$ NMR spectroscopy (see Supplementary Discussion), it was found that cobalt hydride, a key intermediate in the proposed catalytic cycle (Fig. 2), also promoted the oligomerization of styrene[53]. Due to this competing side reaction, 10 equiv of styrene is needed to achieve a high yield of the desired product. Nonetheless, considering the abundance of alkene feedstocks ($ 0.002/mmol for styrene) and the versatile reactivity of olefins, this method is still highly valuable. Hydrogen gas was produced in 64% yield by GC analysis of the crude product mixture (Supplementary Figs. 3–5), which supported our mechanistic hypothesis. The possibility of styrene or styrene oligomers serving as hydrogen acceptors[54] was excluded since byproducts from olefin hydrogenation were not detected (Supplementary Figs. 6–9). During the reaction optimization, we found that the ligand had a large impact on the dehydrogenative alkenylation (Table 1, entries 3–12). Axial ligands on cobaloxime can readily dissociate, and recoordination of these ligands facilitates $\beta$-H elimination[55,56]. Moreover, the electronic and steric properties of axial ligands could dramatically influence the reactivity of cobaloxime complexes[56,57]. A survey of heterocyclic ligands (Supplementary Table 8) revealed that 2,6-lutidine (**L5**) was most effective among all the pyridine ligands evaluated, most probably because it provided a suitable steric environment around the cobalt center. In addition, the use of 50 mol% of 2-methylbenzimidazole (**L9**) was also effective. The use of inorganic bases such as sodium bicarbonate failed to increase reaction yield, confirming that these heterocycles function as ligands rather than bases (Table 1, entry 13 and Supplementary Table 9). No product was detected in the absence of TBADT **9**, cobaloxime **10** or light (Table 1, entry 14), demonstrating that all these components are required. It was noted that other direct HAT photocatalysts resulted in very low conversions under the established conditions (Table 1, entry 15).

**Substrate scope.** With the optimized conditions in hand, we set out to examine the scope and site-selectivity of this C–H alkenylation (Fig. 3). Throughout these studies, the C–H substrate was used as the limiting reagent. A wide range of alkanes were smoothly alkenylated in moderate to good yields with exclusive $E$ selectivity (**11–32**). Cyclic alkanes with ring sizes ranging from 5

**Table 1 Selected optimization results.**

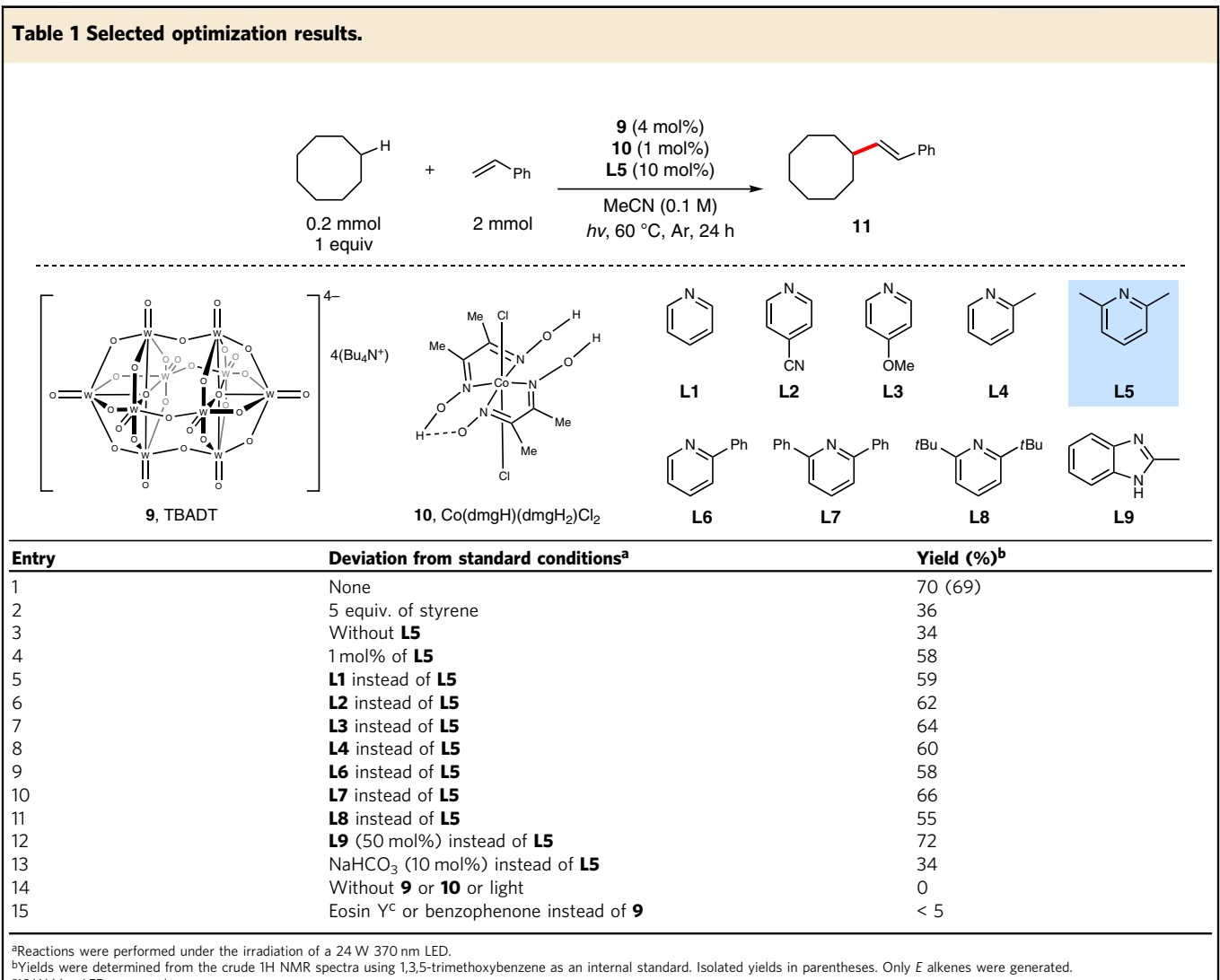

| Entry | Deviation from standard conditions[a] | Yield (%)[b] |
|---|---|---|
| 1 | None | 70 (69) |
| 2 | 5 equiv. of styrene | 36 |
| 3 | Without **L5** | 34 |
| 4 | 1 mol% of **L5** | 58 |
| 5 | **L1** instead of **L5** | 59 |
| 6 | **L2** instead of **L5** | 62 |
| 7 | **L3** instead of **L5** | 64 |
| 8 | **L4** instead of **L5** | 60 |
| 9 | **L6** instead of **L5** | 58 |
| 10 | **L7** instead of **L5** | 66 |
| 11 | **L8** instead of **L5** | 55 |
| 12 | **L9** (50 mol%) instead of **L5** | 72 |
| 13 | NaHCO$_3$ (10 mol%) instead of **L5** | 34 |
| 14 | Without **9** or **10** or light | 0 |
| 15 | Eosin Y[c] or benzophenone instead of **9** | < 5 |

[a]Reactions were performed under the irradiation of a 24 W 370 nm LED.
[b]Yields were determined from the crude 1H NMR spectra using 1,3,5-trimethoxybenzene as an internal standard. Isolated yields in parentheses. Only E alkenes were generated.
[c]18 W blue LED was used.

to 12 gave alkenylated products **11–15** in good yields (54–78%). The alkenylation reaction was amenable to scale-up, delivering alkene **16** in 71% yield on a 5.0-mmol scale. Interestingly, this dual-catalytic system exhibits a strong preference for the functionalization of sterically accessible and electron-rich C–H sites. For instance, the secondary C–H bonds in norbornane (**16**), 1,4-epoxycyclohexane (**17**) and *trans*-1,4-dimethylcyclohexane (**18**) were selectively alkenylated, while the sterically hindered tertiary C–H bonds were not reactive despite their relatively lower bond dissociation energies (BDEs)[58]. Primary C–H bonds, owing to their high BDEs[58], were barely functionalized in the presence of secondary C–H bonds (**18–19**). The steric preference of this transformation is also illustrated by the regiospecific alkenylation of 1-methylcyclopentanol (**19**) at the remote methylene site. Similarly, the least hindered and most electron-rich methylene site of azaspirodecanedione (**20**) was selectively alkenylated. The reaction of adamantane (**21**) furnished two regioisomers (1.3:1) in 82% yield, while the reaction of adamantane derivatives bearing electron-withdrawing groups (**22–24**) occurred predominantly at the tertiary sites. This unique selectivity observed among adamantane derivatives probably originated from the steric accessibility of the tertiary C–H bond due to its equatorial character[59] and the remarkably long life-time of 1-adamantyl radicals compared to 2-adamantyl radicals[60]. For linear alkanes, pentane (**25**, α:β:γ 1.1:4.2:1) and hexane (**26**, α:β:γ 1:3.7:1.7) were preferentially

functionalized at the internal secondary positions. Hexamethyldisilane, which possesses one of the strongest C$_{sp3}$-H bonds (BDE = 101.3 kcal/mol, see Supplementary Discussion), underwent alkenylation to afford product **27** in 50% yield. Activated C$_{sp3}$–H bonds were also examined. Ethers (**28–30**), amides (**31**) and alkylbenzenes (**32**) were all found to be competent substrates (38–67%). Notably, substituted tetrahydropyrans afforded alkenylation products (**29** and **30**) with excellent regio- and diastereoselectivity.

α,β-Unsaturated ketones are core structures in a wide variety of naturally occurring and man-made molecules that are of synthetic, biological and pharmaceutical importance[61]. The direct alkenylation of aldehydes represents a highly attractive and sustainable strategy for the synthesis of this important family of compounds. We were excited to find that various primary and secondary aldehydes were effective substrates under the optimal alkenylation conditions, providing α,β-unsaturated ketones in decent yields with exclusive *E* selectivity (**33–43**). Notably, excellent site selectivity was observed for aldehyde C–H bonds, while other activated C–H bonds, such as benzylic (**34**), allylic (**38**), propargylic (**39**) and α-heteroatom C–H bonds (**42** and **43**), were not functionalized. The preference of formyl C–H abstraction by decatungstate anion can be explained by the polar transition state due to the partial positive charge on the carbonyl carbon (Supplementary Fig. 21)[62]. A tertiary aldehyde (**44**), 2-

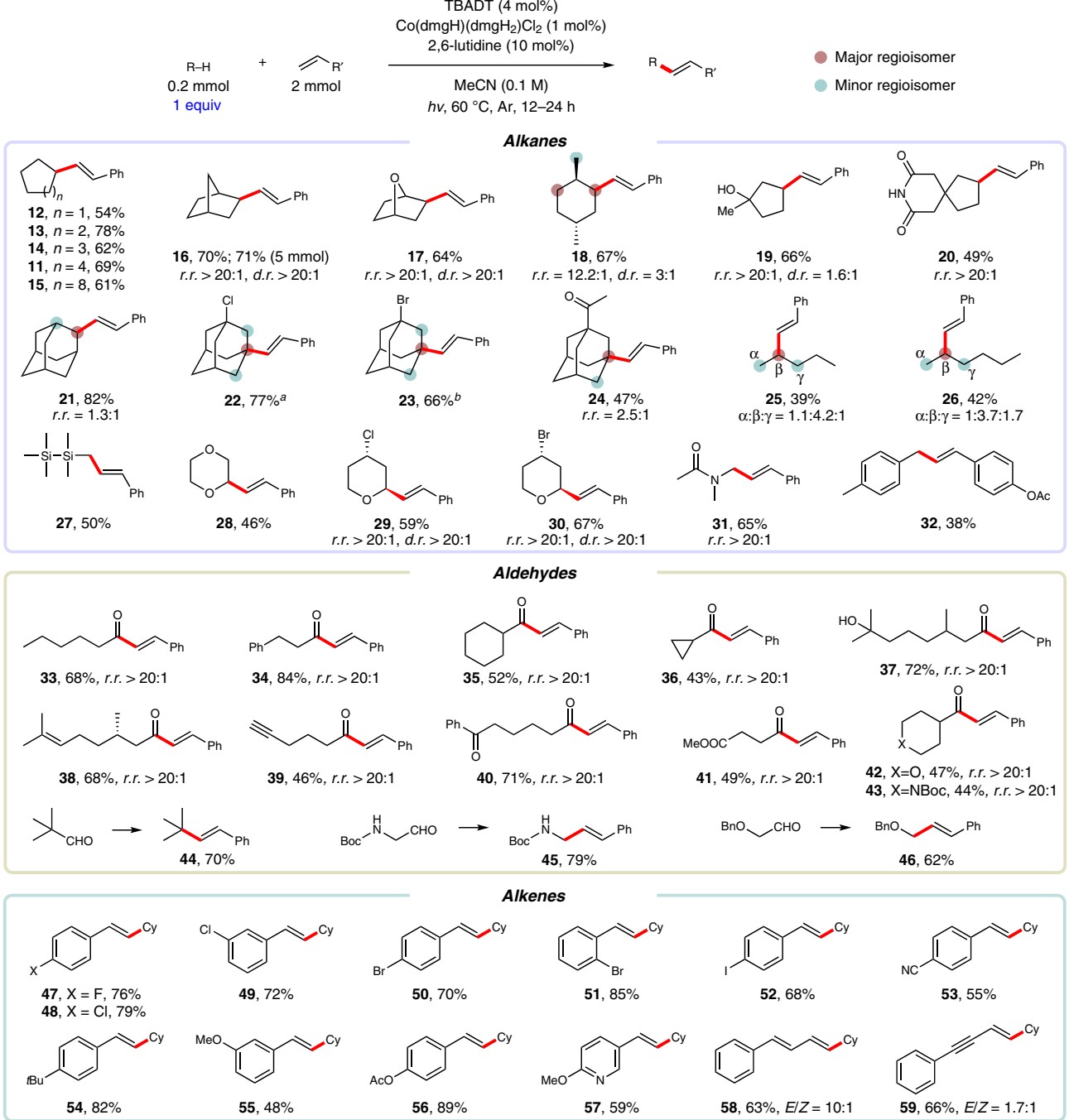

**Fig. 3 Scope of dehydrogenative alkenylation.** Isolated yields. *E/Z* > 99:1 unless otherwise noted. [a]60% selectivity. [b]67% selectivity. Ac = acetyl, Boc = *tert*-butyloxycarbonyl, Bn = benzyl, Cy = cyclohexyl.

*tert*-butyloxycarbonylamino and 2-benzyloxy acetaldehydes (**45** and **46**) acted as alkyl radical equivalents and gave decarbonylative alkenylated products exclusively in high yields (62–79%). However, aromatic aldehydes gave poor yields under the established reaction conditions.

The generality of this C–H alkenylation with respect to the alkene component was subsequently investigated. Electron-deficient (**47–53**) and electron-rich (**54–56**) styrene derivatives possessing *ortho-*, *meta-*, or *para*-substituents were all found to efficiently provide the alkenylated products in moderate to good yields and with exclusive *E* selectivity. The 1,2-disubstituted vinyl pyridine product (**57**) could be obtained in 59% yield. 1,3-Dienes (**58**) and enynes (**59**) were also suitable coupling partners, even

though the *E/Z* selectivity was diminished. Importantly, functional groups that are typically sensitive to transition-metal catalysis, such as alkyl bromides (**23** and **30**), alkenes (**38**), alkynes (**39**) and aryl iodides (**52**), were all well tolerated, allowing for subsequent orthogonal functionalization reactions.

**Late-stage selective C–H alkenylation of complex molecules**. The potential of this protocol for the late-stage site-selective alkenylation of complex molecules was subsequently investigated. As illustrated in Fig. 4, a number of natural products and derivatives were successfully alkenylated under the optimal conditions. The observed regioselectivities were in accordance with the

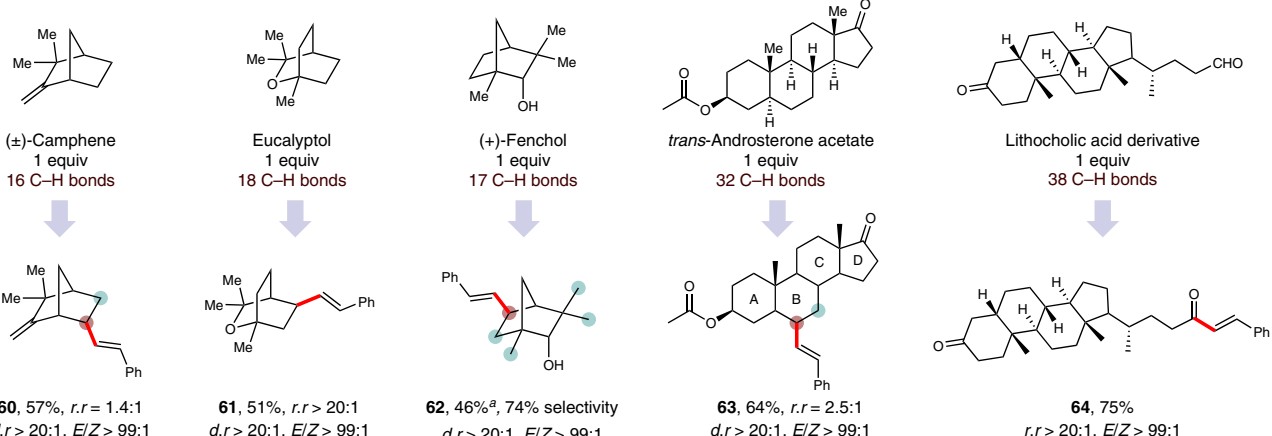

**Fig. 4 Late-stage alkenylation of natural products and derivatives.** Isolated yields. Standard conditions unless otherwise noted. [a]Overall yields over three steps: (1) protection: hexamethyldisilazane (0.6 equiv.), ammonium thiocyanate (0.05 equiv.), DCM, 0 °C → RT, 12 h; (2) alkenylation: standard conditions for dehydrogenative alkenylation; (3) deprotection: tetra-n-butylammonium fluoride (1 equiv.), THF, RT, 12 h. DCM = dichloromethane. THF = tetrahydrofuran.

preferences described above. Camphene, a terminal-olefin-containing natural product, was alkenylated at sterically accessible methylene sites (**60**, 57%). A moderate yield of alkenylated eucalyptol was obtained (**61**, 51%) with excellent site selectivity (over 20 to 1) for the less hindered methylene site. This selectivity is remarkable, especially compared to TBADT-catalyzed photo-oxygenation of eucalyptol (2.4 to 1 methylene selectivity)[63]. Useful efficiencies were observed for the terpenoid (+)-fenchol with 74% regioselectivity for the unactivated and less hindered methylene sites (**62**, 46%). The steroid trans-androsterone acetate, which contains five tertiary C–H bonds and nine methylene sites, underwent selective alkenylation on the B ring (**63**, 64%). This selectivity is predictable, as the C–H bonds on the A ring and D ring are electronically deactivated, while the C–H bonds on the C ring are sterically inaccessible. Finally, a complex aldehyde derivative of lithocholic acid was readily functionalized, delivering α,β-unsaturated ketone **64** in 75% yield with excellent selectivity.

**Mechanistic considerations.** Although excited decatungstate anions have been applied to abstract electron-rich and sterically accessible C–H bonds[64], the excellent sterically controlled selectivity of this C–H alkenylation (e.g., eucalyptol **61**) prompted us to investigate the potential role of the cobaloxime catalysts. Control experiments with different ligands on the cobalt catalyst (Supplementary Table 12) suggested that planar dimethylglyoximate ligands gave the cobalt catalyst exceptional steric bulk[65] and contributed to the high regioselectivity. In addition, various control experiments have been conducted to further support the proposed mechanism in Fig. 2 (see Supplementary Discussion). The presence of radical intermediates was evidenced by radical trapping experiments using 2,2,6,6-tetramethylpiperidine-N-oxyl (TEMPO, Supplementary Fig. 10). Light on-off experiments demonstrated that continuous light irradiation was necessary for the transformation (Supplementary Fig. 13). We measured the kinetic isotope effect (KIE) from two parallel reactions and an intermolecular competition experiment, and the KIE values were calculated to be 1.0 and 2.0 respectively (Supplementary Figs. 14–16). These results indicated that C–H cleavage may not be the rate-determining step[66]. The UV–Vis absorption of the reaction mixture was also monitored. Two absorption bands at 440–500 nm and 550–700 nm were observed after 10 min light irradiation and subsequent exposure to air (Supplementary Figs. 17–18), which indicates the presence of Co$^{II}$ and Co$^{I}$ intermediates respectively[48,49].

## Discussion

In summary, we disclosed a general strategy for the dehydrogenative alkenylation of alkanes and aliphatic aldehydes with aryl alkenes using the C–H substrates as the limiting reagent. The late-stage functionalization of complex pharmaceutically important molecules was achieved with high levels of site selectivity, and a broad scope of functional groups were tolerated, providing a unique method for olefin synthesis and molecular diversification. The sterically and electronically dictated site selectivity originates from the dual effect of both the decatungstate and cobaloxime catalysts. Our future efforts will be directed toward further expanding the alkene scope and developing strategies to further tune the site-selectivity of the C–H alkenylation.

## Methods

**General procedure for dehydrogenative alkenylation of alkanes and aldehydes with alkenes.** To a 10 mL oven-dried Schlenk tube equipped with a magnetic stir bar was added the corresponding C–H nucleophile (0.2 mmol, 1.0 equiv.), alkene (2.0 mmol, 10 equiv.), TBADT (26.6 mg, 0.008 mmol, 4 mol%), Co(dmgH) (dmgH$_2$)Cl$_2$ (0.7 mg, 0.002 mmol, 1 mol%), 2,6-lutidine (2.1 mg, 0.02 mmol, 10 mol %) and dry acetonitrile (2 mL). The resulting mixture was cooled to 0 °C using an ice-water bath, and bubbled with argon balloon for 10 min (if the C–H nucleophile was volatile, it was added after argon bubbling). After that, the reaction was placed under a 370 nm LED (2.5 meter strips, 24 W), stirred and irradiated under argon atmosphere. The temperature was maintained at 60 °C using a water bath. The reaction mixture was removed from light and quenched by stirring open to air for 5 minutes. The solvent was removed on a rotary evaporator under reduced pressure and the residue was subjected to column chromatography isolation over silica gel or preparative thin layer chromatography to obtain the corresponding product.

## Data availability

The authors declare that all other data supporting the findings of this study are available within the article and Supplementary Information files, and also are available from the corresponding author upon reasonable request.

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

## Acknowledgements

We are grateful for the financial support provided by the National University of Singapore and the Ministry of Education (MOE) of Singapore (MOE2017-T2-2-081), NUS (Suzhou) Research Institute and National Natural Science Foundation of China (Grant No. 21702142, 21771135, 21871205). We thank Mr. Tang Jie (National University of Singapore) for the help on quantitative analysis of generated hydrogen gas.

## Author contributions

H.C. discovered and developed the reaction. H.C., Y.K., and J.W. conceived and designed the investigations. X.S. conducted density functional theory (DFT) calculations. H.C., K.L.W., B.B.T., and J.M.C.K. performed the experiments. H.C., X.L., and J.W. wrote the manuscript.

## Competing interests

The authors declare no competing interests.
