## [Peer Review File · Nature Communications]

Reviewers' comments:

Reviewer #1 (Remarks to the Author):

The group of Prof. Wu and coworkers submitted an article dealing with the use of a dual catalytic system for the selective alkenylation of C-H bonds present in several organic compounds including alkanes and aldehydes. The work used a well-known HAT photocatalyst (TBADT) in combination with a Co-based catalyst. Such a combination has been previously used by other groups but NOT for alkenylations (ref. 36).

The reaction has been applied to several compounds, the yields and the selectivity is satisfactory in most cases. The main drawbacks of the work are the use of a large excess of olefins and the exclusive adoption of styrenes as radical acceptors.

Nevertheless, in my opinion the ms could be accepted by the Nat. Comm. journal. I added some requests in order to improve the quality of the work:

- The title is misleading since the alkenylation is not only applied to alkanes and aldehydes.
- In the text the Authors claimed that the C-H cleavage did not occur on "sterically hindered tertiary C-H bonds" but later they claimed that the cleavage "occurred predominantly at the tertiary sites". This contradictory fact needs a comment.
- In the SI the effect of the temperature is apparent. May the Authors can explain why a T higher than r_t is required ?
- The reaction was applied only to styrenes. What about another type of olefins ? A couple of experiments on different class of radical traps should be attempted.
- I was intrigued by the experiments on the synthesis of compounds 35, 42 and 43 starting from secondary aldehydes. In these cases, the resulting acyl radical could be subjected to decarbonylation depending on the T used (see ref. 39). Thus, a high T should favoured the decarbonylation (as in the tertiary aldehyde giving 44) and at least a mixture of carbonylated and decarbonylated derivatives should resulted. How the Authors explained such selective formation of the enone ?
- As for the apparatus, albeit in one case the reaction scale was extended to 5 mmol, I wonder if a simpler system (with no need of a freeze-pump-thaw as an example) could be likewise effective.
- The Authors made some DFT calculations that are not commented in the text (only in one reference). Are these calculations important for this work ?

Reviewer #2 (Remarks to the Author):

Prof. Wu and co-workers described a photocatalytic method for the direct alkenylation of alkanes and aldehydes with aryl alkenes in the absence of any external oxidant. A diverse range of compounds were smoothly alkenylated in useful yields with the C-H partner as the limiting reagent. This method is very important and useful. It should be published soon after addressing the below issues.

1. In the abstract about the description of "the first example of an effective combination of the direct hydrogen atom transfer photocatalysis with cobaloxime-mediated hydrogen-evolution cross-couplings", "an effective combination of the direct hydrogen atom transfer photocatalysis with cobaloxime-mediated" is better to be deleted. Although the combination was not used in hydrogen-evolution cross-couplings, the combination was used for acceptorless dehydrogenation, such as Nat. Commun. 6:10093 doi: 10.1038/ncomms10093 (2015). The highlight of the combination in figure 1C is better to be deleted. In figure 1C, too much advantages of this transformation were highlighted. Some important advantages should be given to better understand the novelty of the manuscript.

2. To avoid any mistake caused by the error of GCMS and ^1H NMR, the direct evidence of H_2 yield should be given at least under the optimized conditions to confirm the oxidant free transformation. In SI, in part V, about "Elucidation of Whether Styrene Serves as a Hydrogen Acceptor", although

H₂ was observed in GC, the yield of hydrogen was not given. From figure 5-figure 8, the author tried to explain the yield of byproducts from olefin hydrogenation was trace. The consumed amount of styrene was determined as 100% (29%+7%+42%+22%) by four components by GCMS and ¹H NMR. What is the error for these four components? If there is 1% error for each four component, the consumed amount of styrene for ethylbenzene is 4% possible? If it is possible, one of important novelty in these transformation, "the oxidant free" should be given carefully.

3. In line of 111, "The use of inorganic bases such as sodium bicarbonate failed to increase reaction yield, confirming the role of ligands". Is it possible to give more discussion of ligand role in the transformation?

4. In the Mechanistic considerations, "In addition, various control experiments including trapping of radical intermediates, trapping of possible benzylic cation intermediates, light on/off experiments, measurement of the kinetic isotope effect, and UV-Vis absorption studies have been conducted to further support the proposed mechanism". As these experiments was shown in SI. It should be give more description for these experiment result and the corresponding conclusion. Else, it is difficult to understand these control experiments.

Response to Reviewer # 1

Comment 1: The group of Prof. Wu and coworkers submitted an article dealing with the use of a dual catalytic system for the selective alkenylation of C-H bonds present in several organic compounds including alkanes and aldehydes. The work used a well-known HAT photocatalyst (TBADT) in combination with a Co-based catalyst. Such a combination has been previously used by other groups but NOT for alkenylations (ref. 36).

The reaction has been applied to several compounds, the yields and the selectivity is satisfactory in most cases. The main drawbacks of the work are the use of a large excess of olefins and the exclusive adoption of styrenes as radical acceptors.

Nevertheless, in my opinion the ms could be accepted by the Nat. Comm. journal. I added some requests in order to improve the quality of the work:

Response to Comment 1: We thank reviewer #1 for all the supportive comments and insightful suggestions.

Comment 2: The title is misleading since the alkenylation is not only applied to alkanes and aldehydes.

Response to Comment 2: We use the word “alkanes” to describe both simple hydrocarbons and functionalized molecules containing C(sp³)-H bonds. The groups of Alexanian (J. Am. Chem. Soc. 2016, 138, 13854-13857), Doyle (J. Am. Chem. Soc. 2018, 140, 14059-14063) and others also describe hydrocarbons and complex molecules as “alkane(s)” in their titles.

Comment 3: In the text the Authors claimed that the C-H cleavage did not occur on “sterically hindered tertiary C-H bonds” but later they claimed that the cleavage “occurred predominantly at the tertiary sites”. This contradictory fact needs a comment.

Response to Comment 3: We thank reviewer #1 for pointing this out. We did observe a preference of secondary C-H bonds over sterically hindered tertiary C-H bonds in products 16-18 and 60-63. However, for adamantane derivatives bearing electron-withdrawing groups, alkenylation occurred predominantly at the tertiary C-H sites (22-24). As adamantane derivatives are structurally unique (J. Am. Chem. Soc. 1986, 108, 2162-2169), the selectivity probably originates from the steric

accessibility of tertiary C-H bonds in adamantane due to the equatorial character (J. Am. Chem. Soc. 1967, 89, 7127-7129) and the remarkably long life-time of 1-adamantyl radicals compared to 2-adamantyl radicals (J. Am. Chem. Soc. 1972, 94, 1177-1183). These comments and references have been added in the manuscript.

Comment 4: In the SI the effect of the temperature is apparent. May the Authors can explain why a T higher than rt is required?

Response to Comment 4: It has been reported that besides photolysis, elevated temperature such as 60 °C could also provide the activation energy to promote C-Co bond homolysis (J. Am. Chem. Soc. 2014, 136, 16788-16791; Dalton Trans. 2014, 43, 4295), an essential step for the formal β -H elimination and alkene formation (J. Am. Chem. Soc. 1992, 114, 10440-10445). At room temperature a low reaction efficiency was observed, and the yield of product was not improved by extending the reaction time. We speculate that the higher temperature will promote the reaction efficiency by accelerating the β -H elimination step.

Comment 5: The reaction was applied only to styrenes. What about another type of olefins? A couple of experiments on different class of radical traps should be attempted.

Response to Comment 5: Besides styrenes, we have also tried other alkenes as radical traps. The reactions of alkanes with electron-deficient alkenes such as benzyl acrylate, vinyl pyridine and vinyl boronic acid pinacol ester gave mainly alkylation products (Giese reaction) and very small amounts of alkenylated products (< 5%). The reactions of alkanes with electron-rich or unactivated alkenes such as vinyl silanes, 1-vinylpyrrolidin-2-one, phenyl vinyl ether and methylenecyclopentane gave low yield of alkenylated products (< 20%). These experimental results have been added to the Supplementary Information (Supplementary Table 11).

Comment 6: I was intrigued by the experiments on the synthesis of compounds 35, 42 and 43 starting from secondary aldehydes. In these cases, the resulting acyl radical could be subjected to decarbonylation depending on the T used (see ref. 39). Thus, a high T should favoured the decarbonylation (as in the tertiary aldehyde giving 44) and at least a mixture of carbonylated and decarbonylated derivatives should resulted. How the Authors explained such selective formation of the enone?

Response to Comment 6: The acyl radicals derived from secondary aldehydes could undergo decarbonylation at high temperature. As our reaction conditions are mild (60 °C) and an excess amount of styrene exists as radical traps, only traces of decarbonylated products were observed in the syntheses of compounds 35, 42 and 43. The decarbonylation rate constant of the acyl radical derived from a typical secondary aldehyde, 2-ethylpentanal, is reported to be $1.4 \times 10^4 \text{ s}^{-1}$ at room temperature in toluene (Chem. Rev. 1999, 99, 1991-2070), while the rate constant of nucleophilic radical adding to styrene is usually around $2 \times 10^5 \text{ s}^{-1}$ at room temperature (Angew. Chem. Int. Ed. 2001, 40, 1340-1371). The majority of acyl radicals derived from secondary aldehydes may directly add to styrene before decarbonylation under our reaction conditions. As for tertiary aldehyde (44), 2-tert-butyloxycarbonylamino and 2-benzyloxy acetaldehydes (45 and 46), the decarbonylation

rate constants of corresponding acyl radicals are reported to be above $1 \times 10^6 \text{ s}^{-1}$ at room temperature (Chem. Rev. 1999, 99, 1991-2070), which is much faster and is probably the reason they gave decarbonylative alkenylated products exclusively.

Comment 7: As for the apparatus, albeit in one case the reaction scale was extended to 5 mmol, I wonder if a simpler system (with no need of a freeze-pump-thaw as an example) could be likewise effective.

Response to Comment 7: We thank Reviewer #1 for this comment. In entry 8, Supplementary Table 10, we mentioned that argon degassing gave the same efficiency as freeze-pump-thaw in the model reaction. Later on, we found that argon degassing gave same efficiency for other substrates as well. We have changed the methods for the easier set-up in both the manuscript and Supplementary Information.

Comment 8: The Authors made some DFT calculations that are not commented in the text (only in one reference). Are these calculations important for this work?

Response to Comment 8: Some DFT calculations were conducted to reveal the bond dissociation energies (BDEs) of different C-H bonds in substrates (18, 19, 27). These calculations are used to support our analysis. For example, it is generally acknowledged that among primary, secondary and tertiary C-H bonds, primary C-H bonds have highest BDEs while tertiary C-H bonds have lowest BDEs. We don't think these calculations are important for this work, and therefore we did not include them in the manuscript.

Response to Reviewer # 2

Comment 1: Prof. Wu and co-workers described a photocatalytic method for the direct alkenylation of alkanes and aldehydes with aryl alkenes in the absence of any external oxidant. A diverse range of compounds were smoothly alkenylated in useful yields with the C-H partner as the limiting reagent. This method is very important and useful. It should be published soon after addressing the below issues.

Response to Comment 1: We thank reviewer #2 for the supportive comments and valuable suggestions.

Comment 2: In the abstract about the description of "the first example of an effective combination of the direct hydrogen atom transfer photocatalysis with cobaloxime-mediated hydrogen-evolution cross-couplings", "an effective combination of the direct hydrogen atom transfer photocatalysis with cobaloxime-mediated" is better to be deleted. Although the combination was not used in hydrogen-evolution cross-couplings, the combination was used for acceptorless dehydrogenation, such as Nat. Commun. 6:10093 doi: 10.1038/ncomms10093 (2015). The highlight of the combination in figure 1C is better to be deleted. In figure 1C, too much advantages of this

transformation were highlighted. Some important advantages should be given to better understand the novelty of the manuscript.

Response to Comment 2: We have changed “This study represents the first example of an effective combination” to “This strategy relies on the synergistic combination” in the abstract. The highlight of the combination in figure 1C is deleted, and only important advantages are illustrated.

Comment 3: To avoid any mistake caused by the error of GCMS and ¹HNMR, the direct evidence of H₂ yield should be given at least under the optimized conditions to confirm the oxidant free transformation. In SI, in part V, about “Elucidation of Whether Styrene Serves as a Hydrogen Acceptor”, although H₂ was observed in GC, the yield of hydrogen was not given. From figure 5-figure 8, the author tried to explain the yield of byproducts from olefin hydrogenation was trace. The consumed amount of styrene was determined as 100% (29%+7%+42%+22%) by four components by GCMS and ¹HNMR. What is the error for these four components? If there is 1% error for each four component, the consumed amount of styrene for ethylbenzene is 4% possible? If it is possible, one of important novelty in these transformation, “the oxidant free” should be given carefully.

Response to Comment 3: Thank Reviewer #2 for this important comment. The yield of alkenylated product and hydrogen gas in the model reaction was 69% and 64% under optimal conditions, respectively, which suggested the transformation was oxidant-free. This data has been included in the manuscript

There might be experimental error for the amount of these components associated with styrene from supplementary figure 5 to figure 8. However, the amount of ethylbenzene was trace (< 1%) and barely detected by GCMS in all cases.

Based on GC, GCMS and NMR analysis, it is not likely that styrene or the styrene derivative serves as a hydrogen acceptor. This transformation is oxidant-free.

Comment 4: In line of 111, “The use of inorganic bases such as sodium bicarbonate failed to increase reaction yield, confirming the role of ligands”. Is it possible to give more discussion of ligand role in the transformation?

Response to Comment 4: We have changed “confirming the role of ligands” to “confirming that these heterocycles function as ligands rather than bases” to make it clearer. There are discussions about the ligand role in the manuscript: “Axial ligands on cobaloxime can readily dissociate, and recoordination of these ligands facilitates β-H elimination” (ref 56, 57). This is probably the reason 10% of ligand gave higher yield than 1% of ligand. We added some more discussion about the ligand role: “Moreover, the electronic and steric properties of axial ligands could dramatically influence the reactivity of cobaloxime complexes” (ref 57, 58).

Comment 5: In the Mechanistic considerations, “In addition, various control experiments including trapping of radical intermediates, trapping of possible benzylic cation intermediates, light on/off experiments, measurement of the kinetic isotope effect, and UV-Vis absorption studies have been conducted to further support the proposed mechanism”. As these experiments was shown in SI. It

should be given more description for these experiment result and the corresponding conclusion. Else, it is difficult to understand these control experiments.

Response to Comment 5: Thank Reviewer #2 for this comment. We have provided descriptions for these control experiment results in mechanistic elucidation:

“In addition, various control experiments have been conducted to further support the proposed mechanism (Fig. 2).⁶⁸ The presence of radical intermediates was evidenced by radical trapping experiments using 2,2,6,6-tetramethylpiperidine-N-oxyl (TEMPO, Supplementary Fig. 10). Light on-off experiments demonstrated that continuous light irradiation was necessary for the transformation (Supplementary Fig. 13). We measured the kinetic isotope effect (KIE) from two parallel reactions and an intermolecular competition experiment, and the KIE values were calculated to be 1.0 and 2.0 respectively (Supplementary Figs. 14-16). These results indicated that C-H cleavage may not be the rate-determining step.⁶⁹ A UV-Vis monitoring study of the reaction mixture revealed that two absorption bands at 440-500 nm and 550-700 nm appeared after 10 min light irradiation and subsequent exposure to air (Supplementary Figs. 17-18), which was in agreement with the formation of Co^{II} and Co^I intermediates respectively.^{48,49”}